# Visualisation of dCas9 target search in vivo using an open-microscopy framework

Koen J.A. Martens [1,2,7], Sam P.B. van Beljouw[1,7], Simon van der Els[3,4], Jochem N.A. Vink[5], Sander Baas[1], George A. Vogelaar[1], Stan J.J. Brouns [5], Peter van Baarlen [3], Michiel Kleerebezem[3] & Johannes Hohlbein [1,6]

CRISPR-Cas9 is widely used in genomic editing, but the kinetics of target search and its relation to the cellular concentration of Cas9 have remained elusive. Effective target search requires constant screening of the protospacer adjacent motif (PAM) and a 30 ms upper limit for screening was recently found. To further quantify the rapid switching between DNA-bound and freely-diffusing states of dCas9, we developed an open-microscopy framework, the miCube, and introduce Monte-Carlo diffusion distribution analysis (MC-DDA). Our analysis reveals that dCas9 is screening PAMs 40% of the time in Gram-positive *Lactoccous lactis*, averaging 17 ± 4 ms per binding event. Using heterogeneous dCas9 expression, we determine the number of cellular target-containing plasmids and derive the copy number dependent Cas9 cleavage. Furthermore, we show that dCas9 is not irreversibly bound to target sites but can still interfere with plasmid replication. Taken together, our quantitative data facilitates further optimization of the CRISPR-Cas toolbox.

[1] Laboratory of Biophysics, Wageningen University and Research, Stippeneng 4, 6708 WE Wageningen, The Netherlands. [2] Laboratory of Bionanotechnology, Wageningen University and Research, Bornse Weilanden 9, 6708 WG Wageningen, The Netherlands. [3] Host-Microbe Interactomics Group, Animal Sciences, Wageningen University and Research, De Elst 1, 6708 WD Wageningen, The Netherlands. [4] NIZO food research, Kernhemseweg 2, 6718 ZB Ede, The Netherlands. [5] Kavli Institute of Nanoscience, Department of Bionanoscience, Delft University of Technology, Van der Maasweg 9, 2629 HZ Delft, The Netherlands. [6] Microspectroscopy Research Facility, Wageningen University and Research, Stippeneng 4, 6708 WE Wageningen, The Netherlands. [7] These authors contributed equally: Koen J.A. Martens, Sam P.B. van Beljouw. Correspondence and requests for materials should be addressed to J.H. (email: Johannes.Hohlbein@wur.nl)

The discovery of clustered regularly interspaced short palindromic repeats (CRISPR) and CRISPR-associated proteins (Cas) as a microbial defence mechanism triggered an ongoing scientific revolution, as CRISPR-Cas can be adapted to perform sequence-specific DNA modification in prokaryotes, archaea, and eukaryotes[1–4]. *Streptococcus pyogenes* Cas9 is a widely used variant[5] and an endonuclease activity-deficient version, termed dead Cas9 (dCas9), has been used to visualise endogenous genomic loci in living cells[6]. The biochemical interaction mechanisms of Cas9 are well understood[7–12]. The DNA-binding protein domain probes the DNA for a specific protospacer adjacent motif (PAM; 5'-NGG-3') via a combination of 3-dimensional diffusion and 1-dimensional sliding on the DNA[9]. Upon recognition of the PAM, the enzyme starts unwinding the DNA double helix to test for complementarity with a 20 nucleotide-long single guide RNA (sgRNA; R-loop formation). If full complementarity is found, Cas9 continues to cleave the DNA at a fixed position 3 nucleotides upstream of the PAM[13].

Optimization of Cas9-mediated genomic engineering in a desired incubation time whilst minimizing off-target DNA cleavage requires exact kinetic information. In the Gram-negative bacterium *E. coli*, an upper limit for the binding time (30 ms) of dCas9 with DNA has been determined in vivo[14], but it is unknown if such binding times are ubiquitous in prokaryotes. In addition, there is a limited understanding of the spatiotemporal relationship between cellular copy numbers of Cas9 proteins, the number of DNA target sites and the duration and dissociation mechanisms of target-bound dCas9. Since genomic engineering of food-related microbes such as Gram-positive lactic acid bacteria[15] is becoming increasingly valuable[16,17], it is important to assess whether previously determined dCas9 kinetic information can be transferred to food-related microbes.

To study the behaviour of dCas9 in vivo with millisecond time resolution, we used single-particle tracking photo-activated localisation microscopy (sptPALM)[18,19]. In sptPALM, a photo-activatable fluorescent protein, which is by default not fluorescently active but can be activated via irradiation, is fused to the protein of interest, and the fusion protein is expressed in living cells. By stochastically activating a subset of the available chromophores, the signal of a single emitter is localized with high precision (~30–40 nm[20,21]) and, by monitoring its position over time, the movement of the protein fusion is followed and analysed[22].

However, sptPALM mostly provides quantitative information if the protein of interest remains in a single diffusional state for the duration of a track (e.g. >40 ms using at least 4 camera frames of 10 ms). As this temporal resolution is insufficient to elucidate in vivo Cas9 dynamic behaviour (<30 ms)[14], we developed a Monte-Carlo based variant of diffusion distribution analysis (MC-DDA, for analytical DDA see ref. [23]) to extract dynamic information on a timescale shorter than the duration of a single track.

In the experimental realisation, we refine existing single-molecule microscopy frameworks and introduce a new design, the miCube. The miCube is constructed from readily available and custom-made parts, ensuring accessibility for interested laboratories. We then use MC-DDA in combination with the miCube in an assay that employs a heterogeneous expression system in order to explore the dynamic nature of DNA-dCas9 interactions in live bacteria and their dependency on (d)Cas9 protein copy numbers. In particular, we assess dCas9 fused to photo-activatable fluorophore PAmCherry2 in the lactic acid bacterium *L. lactis*, in the presence or absence of DNA targets. With this assay, we show that dCas9 is screening PAMs 40% of the time, with each binding event having an average duration of 17 ± 4 ms. Moreover, we show a dependency of bound dCas9 fraction on DNA target-binding sites, which allows quantification of plasmid copy numbers. This, in turn, indicates that bound dCas9 interferes with plasmid replication. These results are combined in a model that predicts Cas9 cleavage efficiencies in prokaryotes.

## Results

**Elucidation of sub 30 ms dynamic interactions with sptPALM.** In the absence of cellular target sites, dCas9 is expected to be present in either one of two states (Fig. 1a): bound to DNA (red), which results in low diffusion coefficients (~0.2 μm²/s); or freely diffusing in the cytoplasm (yellow), which results in high diffusion coefficients (~2.2 μm²/s). If the transitioning between these states is slow compared to the length of each track (here: 40 ms), diffusion coefficient histograms can be fitted with two static states (Fig. 1b, top, Supplementary Fig. 1).

However, if transitioning between the states is on a similar or shorter timescale as the length of sptPALM tracks, these transient interactions of dCas9 with DNA (orange) will result in temporal averaging of the diffusion coefficient obtained from a single track. Therefore, we developed a Monte-Carlo diffusion distribution analysis (MC-DDA; Fig. 1b, bottom, Methods, with an analytical approach available elsewhere[23]) that used the shape of the histogram of diffusion coefficients to infer transitioning rates between diffusional states. The analysis is based on similar approaches used to describe dynamic conformational changes observed with single molecule Förster resonance energy transfer[24–26]. Briefly, MC-DDA consists of simulating the movement and potential interactions of dCas9 inside a cell with a Monte-Carlo approach: the simulated protein is capable of interchanging between interacting with DNA and diffusing freely, defined by $k_{bound→free}$ and $k_{free→bound}$. The MC-DDA diffusional data is compared with the experimental data, and by iterating on the kinetic rates and diffusion coefficients, a best fit is obtained.

**miCube: an open framework for single-molecule microscopy.** For MC-DDA to deduce high kinetic rates, experimental data with high spatiotemporal resolution (< ~50 nm, < ~20 ms) is required. This is challenging, as individual fluorescent proteins have a limited photon budget (<500 photons[27]), and background fluorescence is introduced by the living cells in which the fluorescent proteins are embedded. While suitable commercial microscopes are available, they often lack accessibility or are prohibitively expensive. This has led to the creation of a plethora of custom-built microscopes in the recent past[28–38], ranging from simplified super-resolution microscopes[30–34] to additions to commercial microscopes[35] or extremely low-cost microscopes[36,37].

To increase the accessibility of single-molecule microscopy with high spatiotemporal resolution further, we developed the miCube, an open-source, modular and versatile super-resolution microscope, and provide details to allow interested researchers to build their own miCube or a derivative instrument (Fig. 1c, Supplementary Fig. 2, Methods, https://HohlbeinLab.github.io/miCube). We used 3D-printed components where possible, surrounding a custom aluminium body to minimize thermal drift and provide rigidity. All custom components are supported by technical drawings (Supplementary Figs. 10–18), along with STL files for direct 3D printing. We provide full details on the chosen commercial components, such as lenses, mirrors, and the camera. A detailed description on building a functioning miCube, along with rationale of the design choices, is given in the Methods section. Moreover, we discuss additional options for replacing expensive components with cheaper options.

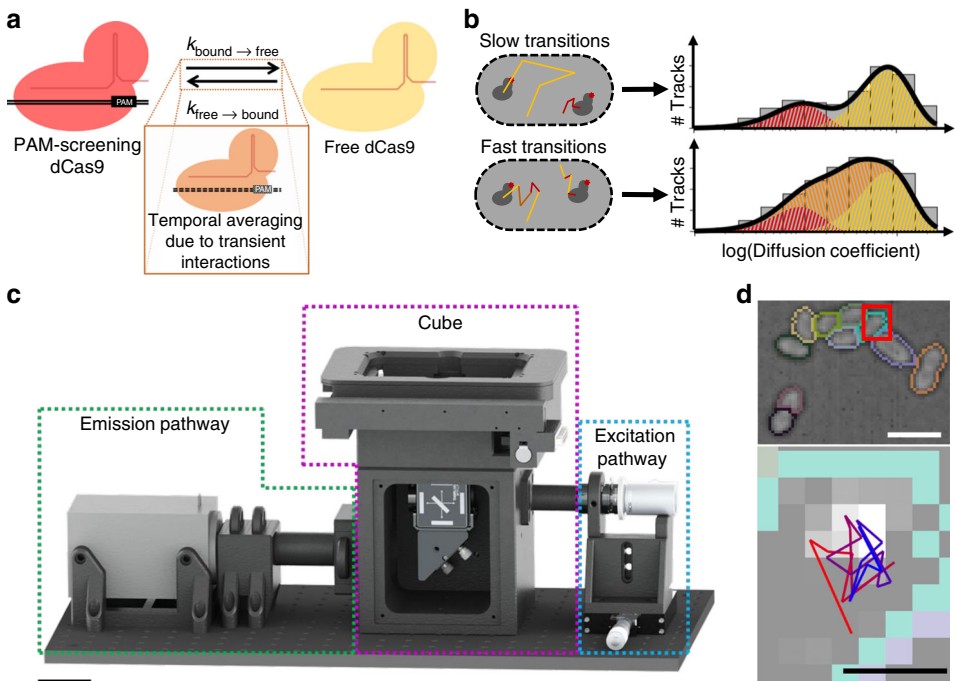

**Fig. 1** Probing cellular dynamics of dCas9 on an open-source microscope using sptPALM. **a** Simplified expected dynamic behaviour of dCas9 in absence of DNA target sites. The protein can be temporarily bound to DNA (PAM screening), or diffuse freely in cytoplasm, with two kinetic rates governing the dynamics. If the interaction is on a similar timescale as the detection time, a temporal averaging due to transient interactions is expected. **b** If the dynamic transitions are slow with respect to the camera frame time used in sptPALM, the obtained diffusional data can be fitted with a static model (top), which assumes that every protein is either free (yellow) or DNA-bound (red), but does not interchange. If the dynamic transitions are as fast or faster than the frame time used, Monte-Carlo diffusion distribution analysis (MC-DDA; bottom) can fit the diffusional data. In MC-DDA, dCas9 can interchange between the two states, resulting in a broader distribution. **c** Render of the open-source miCube super-resolution microscope. The excitation components, main cube, and emission components are indicated in blue, magenta, and green, respectively. Details are provided in the "Methods" section. Scale bar represents 5 cm. **d** Brightfield images of *L. lactis* used for computationally obtaining the outline of the cells via watershed (top), and raw single molecule data (bottom; red outline in top is magnified) as obtained on the miCube as part of a typical experiment, overlaid with the determined track where this single molecule belongs to (starting at red, ending at blue). Scale bars represent 2.5 µm (top) or 500 nm (bottom)

To facilitate straightforward installation and flexible usability of the miCube, we simplified the alignment of the excitation module by decoupling the movement in the three spatial dimensions (Supplementary Fig. 2e). A variety of imaging modalities are possible on the miCube; super-resolution microscopy in 2D and 3D[39], total internal reflection fluorescence (TIRF) microscopy, and LED-based brightfield microscopy. In its current version, the sample area fits a 96-wells plate. The excitation and illumination pathways of the microscope are fitted with 3D-printed enclosures, allowing the instrument to be used under ambient light conditions (including single-particle microscopy). Lastly, we restrained the footprint of the microscope to a $600 \times 300$ mm breadboard (excluding lasers; Supplementary Fig. 2b), further improving accessibility.

Linear drift calculations indicate that the system experiences a drift of $13 \pm 12$ nm/min in the lateral plane and $25 \pm 15$ nm/min in the axial plane without active drift-suppressions systems in place[40] (average of three super-resolution measurements performed on three different days). A typical drift measurement is shown in Supplementary Fig. 3.

**In vivo sptPALM in *L. lactis* on the miCube**. For our sptPALM assay[41], we introduced dCas9 fused to the photo-activatable fluorophore PAmCherry2[27] in *L. lactis* under control of the inducible and heterogeneous *nisA* promotor[42] (pLAB-dCas9, Methods). On the same plasmid, a sgRNA with no fully matching targets in the genome is constitutively expressed. We immobilized the *L. lactis* cells on agarose, and using diffused brightfield LED illumination we computationally separated the cells via the ImageJ watershed[43] plugin (Fig. 1d top). Single-particle microscopy was performed with low induction levels (0.1 ng/mL nisin) and low activation intensities (3–620 µW/cm², 405 nm) to obtain on average PAmCherry2 activation of <1 fluorophore/frame/cell to avoid overlapping tracks (Fig. 1d, bottom). Single particle tracks were limited to individual cells by using the previously obtained cell outlines.

**dCas9 is PAM-screening for 17 ms**. We first assessed the diffusional behaviour of dCas9-PAmCherry2 (hereafter described as dCas9, unless specifically mentioned) in *L. lactis* in the absence of target sites (pNonTarget plasmid; Methods). Under these conditions, dCas9 is expected to diffuse freely in the cytoplasm and screen PAM sites on the DNA for under 30 ms[14]. Under this assumption, diffusion ranges from completely immobile (and thereby fully determined by the localization uncertainty: ~40 nm leads to ~0.16 µm²/s) to freely-moving. The expected free-moving diffusion coefficient can be theoretically described: the fusion protein has a hydrodynamic radius of 5–6 nm[27,44], resulting in a diffusion coefficient of 36–43 µm²/s[45]. Cytoplasmic retardation of ~20× due to increased viscosity and crowding effects reduces this to ~1.8–2.2 µm²/s[46]. We obtained diffusion coefficients in the range of ~0–3 µm²/s (Fig. 2a), which is within the expected range.

We used a heterogeneous promotor (*nisA*, Methods), causing the apparent cellular dCas9 copy numbers to vary between 20 and ~1000 (Fig. 2a, Supplementary Fig. 4; cells with less than 20 copies were excluded as we corrected for ~7 tracks (~14 apparent

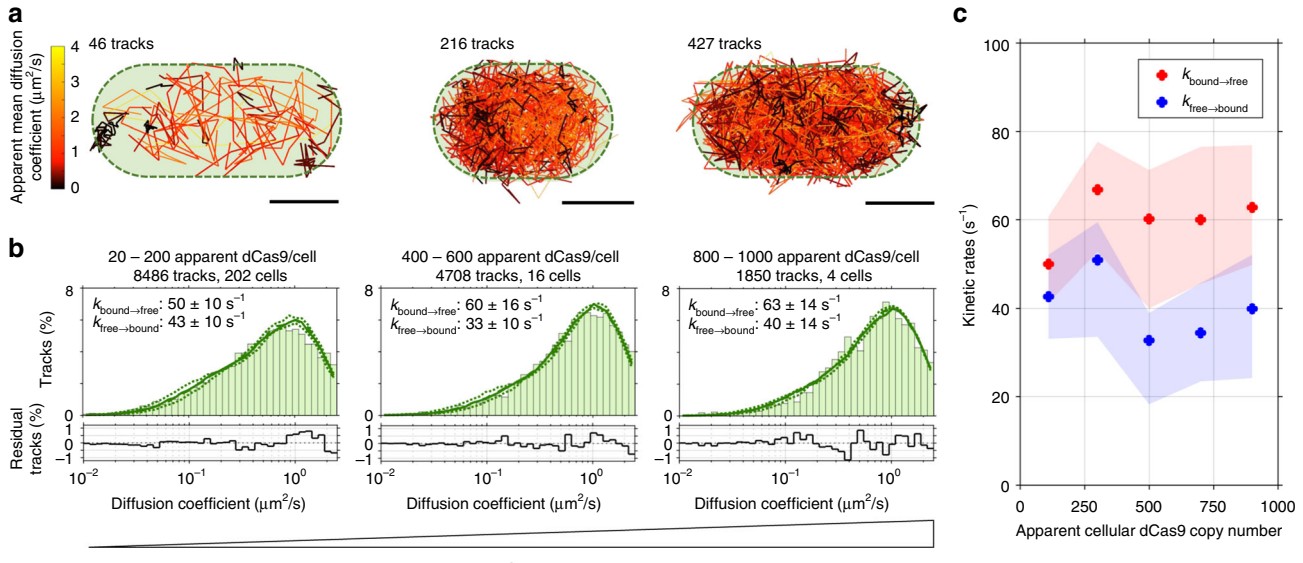

**Fig. 2** sptPALM of dCas9-PAmCherry2 in pNonTarget *L. lactis* with increasing dCas9 concentration. **a** Identified tracks in single pNonTarget *L. lactis* cells. Tracks are colour-coded based on their diffusion coefficient. Three separate cells are shown with increasing cellular concentration of dCas9. Green dotted outline is an indication for the cell membrane. Scale bars represent 500 nm. **b** Diffusion coefficient histograms (light green) belonging to 20–200, 400–600, and 800–1000 dCas9 copy numbers, from left to right. Histograms are fitted (dark green line) with a theoretical description of state-transitioning particles between a mobile and immobile state (dashed line represents 95% confidence interval based on bootstrapping the original data). Five diffusion coefficient histograms (Supplementary Fig. 4) were globally fitted with a single free diffusion coefficient ($2.0 \pm 0.1\,\mu m^2/s$; mean ± standard deviation), a single value for the localization error ($\sigma = 38 \pm 3\,nm = 0.15 \pm 0.03\,\mu m^2/s$), and 5 sets of $k_{bound \rightarrow free}$ and $k_{free \rightarrow bound}$ values (indicated in the figures). Residuals of the fit are indicated below the respective distribution. **c** $k_{bound \rightarrow free}$ (red) and $k_{free \rightarrow bound}$ (blue) plotted as function of the apparent cellular dCas9 copy number. Solid dots show the fits of the actual data; filled areas indicate the 95% confidence intervals obtained from the bootstrapped iterations of fitted MC-DDAs with 20,000 simulated proteins. Source data are provided as a Source Data file

dCas9) found in non-induced cells). The value of the cellular dCas9 is an approximation (Discussion), but a relative increase in cellular dCas9 copy number is certain. We then created five diffusional histograms belonging to cells with a particular apparent dCas9 copy number range (ranges of ~200 dCas9 copy number intervals; Fig. 2b, Supplementary Fig. 4). These diffusional histograms are fitted with the aforementioned MC-DDA, where the shape of the MC-DDA is governed by the localization uncertainty, the free-moving diffusion coefficient, and the kinetic rates of PAM-screening. The localization uncertainty and free-moving diffusion coefficient are independent of cellular dCas9 copy number, since they are determined by the number of photons and a combination of hydrodynamic radius and cytoplasm viscosity, respectively. Therefore, the histograms were globally fitted with a combination of 5 MC-DDAs, each consisting of 20,000 simulated dCas9 proteins, containing a single value for free-moving diffusion coefficient ($D_{free} = 2.0 \pm 0.1\,\mu m^2/s$ (average ± standard deviation of 4 experiments over 3 days, in total consisting of 32,971 tracks), in agreement with the theoretical expectation of ~1.8–2.2 $\mu m^2/s$), a single value for localization uncertainty ($\sigma = 38 \pm 3\,nm$, or $D_{immobile}* = 0.15 \pm 0.03\,\mu m^2/s$, expected for fluorescent proteins illuminated for 4 ms[39,41]), and five pairs of $k_{free \rightarrow bound}$ and $k_{bound \rightarrow free}$ (specified in Fig. 2b, c).

The obtained kinetic constants of $k_{free \rightarrow bound}$ and $k_{bound \rightarrow free}$ were $40 \pm 12\,s^{-1}$ and $60 \pm 13\,s^{-1}$ (mean ± 95% CI), respectively, and did not show a significant dependence on apparent cellular dCas9 copy number (Fig. 2c). This indicates that dCas9 is PAM-screening for $17 \pm 4\,ms$ in *L. lactis*, consisting of screening 1 or more PAMs via 1D diffusion. This value is in the same order of magnitude as the upper limit of 30 ms reported earlier for PAM-screening in *E. coli*[14], suggesting that

these PAM-screening kinetics are a general feature of dCas9. Additionally, dCas9 is on average diffusing within the cytoplasm for $25 \pm 8\,ms$ before finding a new site for PAM screening. This duration is governed by the diffusion coefficient of the fusion protein, along with the average distance between DNA PAM sites. These results also entail that dCas9 is diffusing in the cytoplasm ~60% of the time, while interacting with the DNA ~40% of the time. Removal of the sgRNA resulted in similar diffusional data, which agrees with PAM-screening being a solely protein–DNA interaction ($k_{free \rightarrow bound}$: $34 \pm 16\,s^{-1}$; $k_{bound \rightarrow free}$: $62 \pm 21\,s^{-1}$; diffusion time on average $29 \pm 18\,ms$; PAM-screening time on average $16 \pm 6\,ms$; Supplementary Fig. 5). This also indicates that partial sgRNA-DNA matching of dCas9 with non-targets is not prevalent enough in our assay to affect the screening time significantly.

**Target-binding of dCas9 can be observed with sptPALM.** We then investigated the effect of DNA target sites complementary to the sgRNA loaded dCas9. To this end, we introduced 5 target sites on a plasmid (pTarget; Methods), which replaced the pNonTarget plasmid used so far. Qualitative visualisation of diffusion in the *L. lactis* bacteria shows tracks with small diffusion coefficients (Fig. 3a, black tracks), indicative of target-bound dCas9. This immobile population can be observed throughout the dCas9 copy number range but is more prevalent in cells with lower cellular dCas9 copy numbers.

We expect target-bound dCas9 to move with a diffusion coefficient determined by the plasmid size, which is independent on the cellular dCas9 copy number. Therefore, we globally fitted the pTarget-obtained diffusional histograms with a combination of the corresponding pNonTarget MC-DDA fit and an additional single diffusional state belonging to target-bound dCas9 (Fig. 3b,

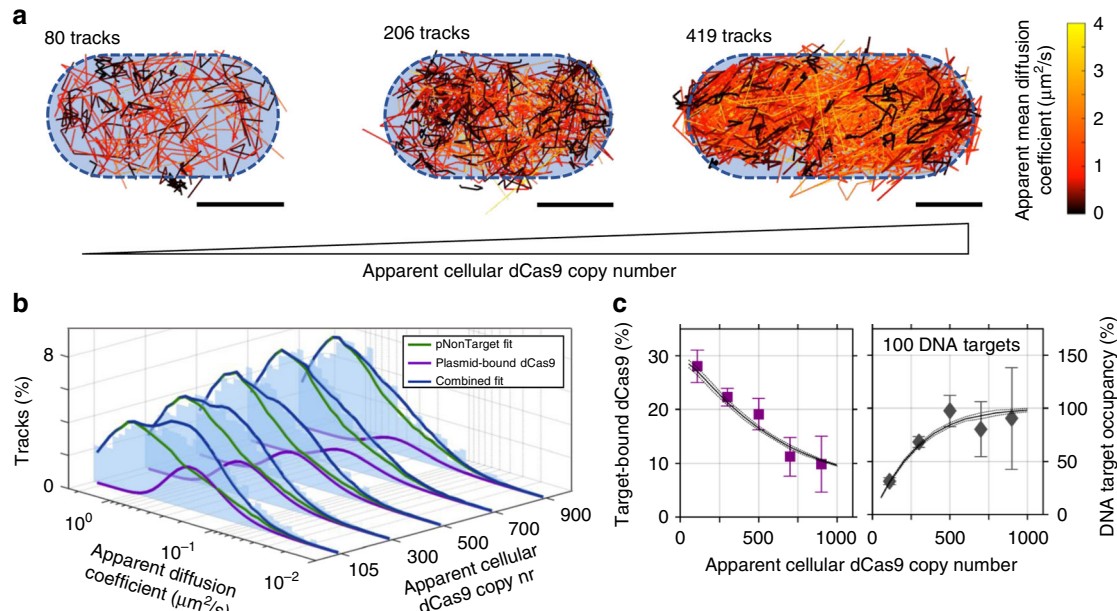

**Fig. 3** sptPALM of dCas9-PAmCherry2 in pTarget *L. lactis* shows target-binding behaviour of dCas9. **a** Identified tracks in individual pTarget *L. lactis* cells. Tracks are colour-coded based on their diffusion coefficient. Three separate cells are shown with increasing dCas9 concentration. Blue dotted outline is an indication for the cell membrane. Scale bars represent 500 nm. **b** Diffusion coefficient histograms (light blue) are fitted (dark blue line) with a combination of the respective fit of pNonTarget *L. lactis* cells (green line), along with a single globally fitted population corresponding to target-bound dCas9 (purple) at $0.38 \pm 0.04$ μm²/s (mean ± standard deviation). **c** Left: The population size of the plasmid-bound dCas9 decreases as a function of the cellular dCas9 copy number. The error bar of the measurement is based on the 95% confidence interval determined by bootstrapping; the solid line is a model fit with 20 plasmids, with a 95% confidence interval determined by repeating the model simulation. Right: Occupancy of DNA targets by dCas9 based on 20 target plasmids (100 DNA target sites), based on the same data as presented in the left figure. Source data are provided as a Source Data file

$D_{plasmid}{}^{\star} = 0.38 \pm 0.04$ μm²/s $= D_{immobile}{}^{\star} + 0.23$ μm²/s, which agrees with the expected diffusion coefficient from plasmids of similar size in bacterial cytoplasm[46–48]; 31,439 total tracks). The plasmid-bound dCas9 population decreases with increasing apparent cellular dCas9 copy numbers from $28 \pm 3\%$ at 105 (20–200) copies to $10 \pm 5\%$ at 900 (800–1000) copies (Fig. 3c left, purple squares; mean ± 95% CI). No target-binding behaviour was observed when the sgRNA was removed (Supplementary Fig. 5).

**dCas9 does not bind targets irreversibly.** This anti-correlation between dCas9 copy number and the size of the plasmid-bound population is indicative of competition for target sites by an increasing amount of dCas9 proteins. To evaluate this hypothesis, we consecutively simulated dCas9 proteins until the cellular dCas9 copy number was reached (Methods). In the simulation, every protein binds or dissociates from a PAM with the kinetic constants determined previously, and will instantly bind to a target site if it binds to a PAM directly adjacent to it. We thus disregard effects of 1D sliding on the DNA, but we believe these effects are limited, as 1D sliding between PAM sites has a low probability when PAMs are randomly positioned on the DNA ($< \sim 10\%$ at 16 bp distance average[9]). A $k_{off}$ is introduced which dictates removal of dCas9 from the target sites.

This model fully explained the dependency of the target-bound dCas9 fraction on the cellular dCas9 copy number (Fig. 3c left, black line). The slope of the curve towards low cellular dCas9 concentration is dependent on the total cellular number of PAM sites and $k_{off}$. Assuming on average 1.5 genomes worth of DNA (haploid genome replicated in half the cells) present in the cell, the $k_{off}$ is $\sim 0.01 \pm 0.003$ s⁻¹. The number of DNA target sites determines the lower bound of the model, and $\sim 100 \pm 50$ DNA target sites ($\sim 20 \pm 10$ plasmids) led to the observed bound

fraction at 900 cellular dCas9 proteins. The fit of the number of target sites at high cellular dCas9 concentration is independent of $k_{off}$, since at the modelled concentrations and PAM-screening kinetic parameters, the target sites are essentially fully occupied (Fig. 3c, right). It thus follows that the used pTarget plasmid, a derivative of pNZ123, is present at a lower copy number than expected ($\sim 60$–80) during measurements[47]. This could hint towards interference of plasmid replication due to dCas9 binding[49,50]. We investigated this with quantitative polymerase chain reaction (qPCR)[51], and we indeed observed a decrease in the amount of pTarget DNA with dCas9 production (Supplementary Fig. 6).

These collective results lead to the model presented in Fig. 4a. dCas9 diffuses freely in the cytoplasm for $25 \pm 8$ ms on average, and will then interact with a PAM site for $17 \pm 4$ ms. If the PAM site is not directly adjacent to a target site, dCas9 will move back to freely diffusing in the cytoplasm. However, if the PAM site is directly followed by a target site, dCas9 will be bound to this site for 1.6 min on average, before it is removed by intrinsic or extrinsic factors.

**A single copy of Cas9 find a single DNA target in ~4 h.** We adapted the computational target-binding model to predict Cas9 cleavage in *L. lactis* and other prokaryotes with similar DNA content. We assume that all DNA is accessible to Cas9 and that Cas9 behaves identical to dCas9, but will cleave a target directly after binding. Our proposed Cas9 kinetic scheme depends only on PAM-screening kinetic rates and the ratio of total PAM sites to target sites. We predicted the incubation time-dependent probability that a certain number of cellular Cas9 proteins will bind a single target site on the *L. lactis* genome (Fig. 4b).

The model shows that a single Cas9 protein can effectively find a single target with 50% probability in ~4 h. It also shows that an

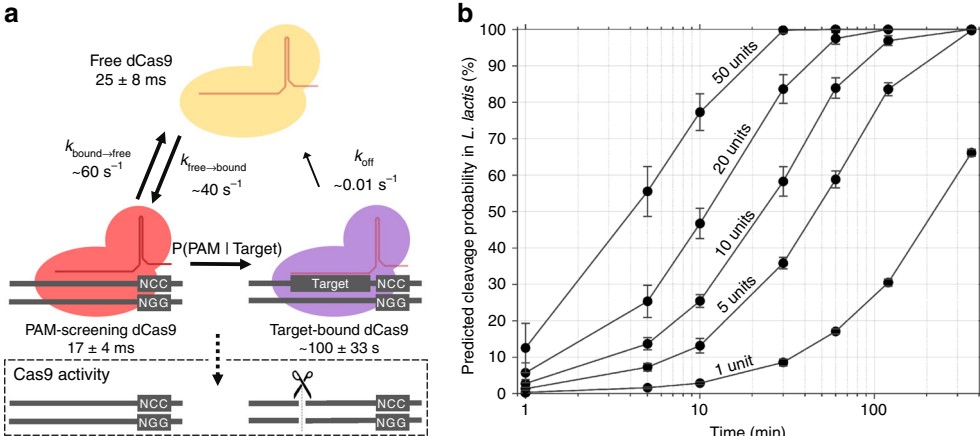

**Fig. 4** Extrapolation of the dCas9 dynamic model to assess single target cleavage by Cas9. **a** The proposed model surrounding dCas9 interaction with the obtained kinetic rates. Free dCas9 (yellow) in the cytoplasm interact with PAM sequences (5′-NGG-3′) on average every 25 ms. If the PAM is not in front of a target sequence (red), only PAM-screening will occur for on average 17 ms. If the PAM happens to be in front of a target, the dCas9 will be target-bound (purple). We extend this model to predict Cas9 cleavage under conditions where target-bound Cas9 will always cleave the target DNA. **b** Calculated predicted probability that a single target in the *L. lactis* genome is cleaved after a certain period of time with a certain cellular Cas9 copy number, based on the model shown in a. Error bars indicate standard deviation calculated from iterations of the model

increasing cellular Cas9 copy number quickly decreases this search time: With 10 cellular copies of Cas9, the search time is reduced to ~25 min, and 20 copies reduce the search time to ~10 min. Therefore, a single target is almost certainly found within a typical prokaryotic cell generation time (> ~20 min). This agrees with in vivo data of Cas9[14] (accounting for *E. coli*'s larger genome (~4.6 mbp versus ~2.5 mbp)) and with in vivo data of Cascade in *E. coli*[23], though in different organisms or with different CRISPR-Cas systems.

## Discussion

We have designed a sptPALM assay to probe DNA-protein interactions in vivo, and assessed the kinetic behaviour of dCas9 in *L. lactis* on the open-hardware, super-resolution microscope miCube. The high spatiotemporal resolution of the experimental data along with the heterogeneity of the used induction protocol allowed us to develop a Monte-Carlo diffusion distribution analysis (MC-DDA) of the diffusional equilibrium.

The obtained dCas9 PAM-screening kinetic rates ($k_{\text{free}\rightarrow\text{bound}} = 40 \pm 12\,\text{s}^{-1}$, $k_{\text{bound}\rightarrow\text{free}} = 60 \pm 13\,\text{s}^{-1}$) indicate that non-target binding of dCas9 has a mean lifetime of $17 \pm 4\,\text{ms}$, and spends ~40% of its time on PAM screening. In fact, a 1:1 ratio between diffusing and binding was shown to be optimal for target search time of DNA-binding proteins[52]. The MC-DDA further suggests that the kinetic rates governing PAM–dCas9 interactions do not depend on cellular copy number levels of dCas9.

We observed target-binding of dCas9, and showed that higher cellular dCas9 copy numbers resulted in lower probabilities of target-bound dCas9, although absolutely more targets were occupied by dCas9. We linked this finding to the previously found $k_{\text{free}\rightarrow\text{bound}}$ and $k_{\text{bound}\rightarrow\text{free}}$ rates and postulate that dCas9 dissociation from target sites is responsible for the obtained probabilities of target binding by dCas9. We made two assumptions when obtaining absolute cellular dCas9 copy numbers. Firstly, we assumed that measurements directly end after all fluorophores in the centre of the microscopy field of view have been imaged once. Secondly, we assumed a maturation grade of 50% (identical to that of PAmCherry1 in *Xenopus*[53]). Although an exact determination is possible[53,54], this is beyond the scope of this study.

We obtained a dCas9-target $k_{\text{off}}$ rate of ~0.01 s$^{-1}$ that is dependent on the exact cellular Cas9 copy number and total

*L. lactis* genomic content. The biological cause of dissociation of target bound dCas9 from DNA remains speculative: it could be an intrinsic property, resulting in spontaneous release from target sites, or it could be caused by an extrinsic factor, such as RNA transcription or DNA replication. We do not expect RNA polymerase activity on the DNA target sites, although we did not actively block transcription. It is currently unknown whether genomic target-bound dCas9 dissociates from the DNA due to DNA replication, with studies contradictory showing that dCas9 is removed during cell duplication[14] and that dCas9 is hindering genomic DNA replication[49] or transcription[50]. We note that genomic DNA replication substantially differs from the rolling-circle DNA replication of pTarget[55].

Our data indicate that dCas9 binding to plasmid DNA hinders DNA rolling-circle replication. The pNZ123 plasmid, of which pTarget is a derivative, is believed to be high-copy[47] (60–80 plasmids per cell), although the quantification of plasmid copy numbers is challenging (discussed for the single-cell level in reference[51]). Our model suggests that pTarget is present in only ~20 copy numbers during our measurements. Although we saw an effect of dCas9 production on pTarget copy number via qPCR, the obtained decrease (~20%) is not as large as observed with sptPALM (~70%). The median cellular dCas9 copy number, however, is low (~40; Supplementary Fig. 6) compared to most of the dCas9 copy number bins evaluated with MC-DDA. Therefore, using the averaged cellular community, not all pTarget (60–80 cellular plasmids containing 300–400 target sites), are occupied by a dCas9 protein, which would affect the ensemble qPCR results. The sptPALM plasmid copy number determination, on the other hand, is mostly determined by the *L. lactis* sub-population with high dCas9 copy numbers, for which pTarget replication is restricted more strongly.

We used our model to make predictions about Cas9 cleavage probabilities, based on kinetic values extracted from the MC-DDA, which are not influenced by the approximated cellular dCas9 copy number. The kinetic parameters of dCas9-PAmCherry2 provide estimates for those of Cas9. We reason that $k_{\text{bound}\rightarrow\text{free}}$ will be unchanged, since this rate is based on the duration of the PAM screening, while $k_{\text{free}\rightarrow\text{bound}}$ will be slightly lower for Cas9 compared to dCas9-PAmCherry2, due to the relatively higher diffusion coefficient of Cas9. The model can be expanded to incorporate a protein diffusion coefficient to obtain a

modified $k_{\text{free}\to\text{bound}}$ rate, and to include accessibility of the DNA. These additions would allow the model to predict Cas9 behaviour in more diverse environments such as eukaryotic cells. Other computational models have taken these parameters into account[56], but these models were not based on experimental in vivo data, and were based on different assumptions.

Our open microscopy framework enables the study of in vivo protein–DNA interactions with high spatiotemporal resolution, here shown for CRISPR-Cas9 target search, and improves the general accessibility of super-resolution microscopy. Our data shows that heterogeneity in an expression system can be used to obtain new insights in any protein–DNA or protein–protein interaction in vivo, here indicating that target-bound dCas9 interferes with rolling-circle DNA replication. The derived kinetic parameters and information on target search times provide valuable practical insights in CRISPR-Cas engineering and gene silencing in lactic acid bacteria specifically, and suggest to reflect prokaryotic Cas9 search times in general.

## Methods

**miCube design considerations.** We designed the miCube to be easy to setup and use, while retaining a high level of versatility. The instrument and its design choices will be described in three parts: the excitation path; the emission path, and the cube connecting the sample with the excitation and emission paths. Throughout this description, we will refer to numbered parts as shown in Supplementary Fig. 2a, c and described in Supplementary Table 1. The information on the miCube presented here can also be found on https://HohlbeinLab.github.io/miCube/component_table.html. The instrument is fully functional in ambient light, due to a fully enclosed sample chamber, illumination pathway and emission pathway. Moreover, the miCube has a small footprint: the final design of the miCube, excluding the lasers and controllers, fits on a $300 \times 600$ mm Thorlabs breadboard. We placed the whole ensemble in a transparent polycarbonate box (MayTec Benelux, Doetinchem, The Netherlands) to minimize airflow disturbing the setup during experiments.

**miCube excitation path.** The excitation path is designed to be both robust and easy to align and adjust. The four laser sources located in an Omicron laser box are combined and guided via a single mode fibre towards a reflective collimator (nr. 18) ensuring a well-collimated beam. The reflective collimator is attached directly to an aperture (nr. 17), a focusing lens (nr. 16, 200 mm focal length), and an empty spacer (nr. 12). This excitation ensemble is placed in the 3D-printed piece designed to hold the assembly into place (nr. 13). This holder is then attached to a right-angled mounting plate (nr. 14), which is placed on a 25 mm translation stage (nr. 15). The translation stage should be placed at such a position on the breadboard that the focusing lens (nr. 16) is exactly 200 mm separated from the back-focal plane of the objective when following the laser path.

Easy alignment and adjustment are ensured by isolating the three axes of movement of this excitation ensemble (Supplementary Fig. 2e). Adjustments of distance from objective is achieved by moving the collimator ensemble (nrs. 12, 16–18) inside its holder (nr. 13). Height of the path can be adjusted via a bracket clamp that supports the collimator ensemble (nrs. 13 and 14), and the horizontal alignment can be adjusted via a translation stage where the bracket clamp rests on (nr. 15). We note that the excitation pathway is uncoupled from any laser source due to the fibre-connection, allowing for freedom of choice for the excitation laser unit.

Additionally, the translation stage (nr. 15) can be used to enable highly inclined illumination (HiLo) or total internal reflection (TIR). The stage allows fine and repeatable adjustment of the excitation beam position on the back focal plane of the objective. By aligning the excitation beam in the centre of the objective, the microscope will act as a standard epifluorescence instrument. If the excitation beam is aligned towards the edge of the back focal plane, the miCube will operate in HiLo or TIR.

**miCube cube and sample mount.** The central component of the miCube is the cube (nr. 5) that connects excitation path, emission path, and the sample. The cube is manufactured out of a solid aluminium block maximising stability and minimising effects of drift due to thermal expansion. Black anodization of the block prevents stray light and unwanted reflections. The illumination path is further protected from ambient light by screwing a 3D-printed cover (nr. 11) on the side of the cube, as well as a door to close the cube off.

Next, the dichroic mirror—full mirror part is assembled (nrs. 6–10). The dichroic mirror unit (nr. 7) consists of a dichroic mount that is magnetically attached to an outer holder. On the side of the dichroic mirror unit, opposing the excitation path, a neutral density filter (nr. 6) is placed to prevent scattered non-reflected light entering the miCube thereby minimizing background signal being recorded by the camera. At the bottom of the dichroic mount assembly, a TIRF filter (nr. 8) is placed to remove scattered back-reflected laser light from entering the emission pathway. This ensembled dichroic mirror unit is screwed via a coupling element (nr. 9) to a mirror holder containing a mirror placed at a 45° angle (nr. 10), which reflects the emission light from the objective to the camera. This completed dichroic mirror—full mirror part is screwed into the backside of the miCube via two M6 screws, which hold the ensemble into place and directly in line with the excitation path (nrs. 12-18), the objective (nr. 3), and the tube lens (nr. 30).

Then, an objective (nr. 3) (Nikon 100× oil, 1.49 NA, HP, SR) is directly screwed into an appropriate thread on top of the cube. Around the objective, a sample mount (nr. 4) is screwed on top of the cube, which is designed to ensure correct height of the sample with respect to the parfocal distance of the chosen objective. We opted for using a sample mount, as it can be easily swapped for another to retain freedom in peripherals. For example, only the height of the sample mount has to be changed if an objective has a different parfocal distance as the one used here. We designed two different sample mounts (nr. 4a, 4b). The first one can hold an $xy$-translation stage with $z$-stage piezo insert (nr. 2), to enable full spatial control of the sample (nr. 4a). The second one only holds the $z$-stage piezo insert, which decreases instrument cost (nr. 4b). In any case, the $xy$-translation stage with $z$-stage piezo insert, or only the $z$-stage piezo insert is screwed in place into corresponding threaded holes in the sample mount. A glass slide holder (nr. 1) is made from aluminium to fit inside a 96-wells-holder like the $z$-stage (nr. 2).

**miCube detection path.** A tube lens ensemble is made (nrs. 27–30) which houses a 200 mm focal length tube lens (Thorlabs) in a 3D-printed enclosure which provides space to slot in an emission filter (nrs. 27, 28). This ensemble is then attached directly to the miCube by screwing it into place with four M6 screws. The alignment of the tube lens is therefore exactly in line with the emission light, as the centre of the full mirror (nr. 10) is at the same height of the tube lens. The direction of the emission light can be aligned, which can simply be achieved by tuning the angle of the full mirror (nr. 10).

A cover (nr. 25) is attached to the tube lens ensemble to ensure darkness of the emission path, which is connected to the tube lens by a 3D-printed connector piece (nr. 26). On the other end of the cover, a 3D-printed holder for 2 astigmatic lenses (nr. 21) is placed and screwed into place in the breadboard. Astigmatic lenses (nrs. 22-24) can optionally be used to enable 3D super-resolution microscopy[57]. They can be easily changed for lenses with a different focal length or with empty holders. With this, astigmatism can be enabled or disabled, and a choice between more accurate $z$-positional information with a lower $z$-range, or less accurate information with a larger range can be made. The Andor Zyla 4.2 PLUS camera (nr. 19) is placed behind the astigmatic lens holder, and is positioned in a 3D-printed camera mount (nr. 20) to ensure correct height and position of the camera, so that the focus of the tube lens is at the camera chip. We chose for a scientific Complementary Metal-Oxide Semiconductor (sCMOS) camera to take advantage of a larger field of view and increased temporal resolution compared to the more traditional electron-multiplying charge coupled device (EMCCD) cameras[58].

Note that the length of the cover (nr. 25) and the alignment of the holes at the feet of the 3D-printed astigmatic lens holder (nr. 21) are dependent on the focal length of the tube lens, and of the position of the chosen camera chip with regards to the 3D-printed mount for the camera. The pieces used here were designed for the Andor Zyla 4.2 PLUS, a 200 mm focal length tube lens, and the used custom-designed camera mount (nr. 20).

**Strain preparation and plasmid construction.** *Lactococcus lactis* NZ9000 was used throughout the study. NZ9000 is a derivative of *L. lactis* MG1363[59] in which the chromosomal *pepN* gene is replaced by the *nisRK* genes that allow the use of the nisin-controlled gene expression system[42]. Cells were grown at 30 °C in GM17 medium (M17 medium (Tritium, Eindhoven, The Netherlands) supplemented with 0.5% (w/v) glucose (Tritium, Eindhoven, The Netherlands) without agitation.

**DNA manipulation and transformation.** Vectors used in this study are listed in Supplementary Table 2. Oligonucleotides (Supplementary Table 3) and primers Supplementary Table 4) were synthesised at Sigma-Aldrich (Zwijndrecht, The Netherlands). Plasmid DNA was isolated and purified using GeneJET Plasmid Prep Kits (Thermo Fisher Scientific, Waltham, MA, USA). Plasmid digestion and ligation were performed with Fast Digest enzymes and T4 ligase respectively, according to the manufacturer's protocol (Thermo Fisher Scientific, Waltham, MA, USA). DNA fragments were purified from agarose gel using the Wizard SV gel and PCR Clean-Up System (Promega, Leiden, The Netherlands). Electro competent *L. lactis* NZ9000 cells were generated using a previously described method[60]. Prior to electro-transformation, ligation mixtures were desalted for one hour by drop dialysis on a 0.025 μm VSWP filter (Merck-Millipore, Billerica, US) floating on MQ water. Electro-transformation was performed with GenePulser XcellTM (Bio-Rad Laboratories, Richmond, California, USA) at 2 kV and 25 μF for 5 ms. Transformants were recovered for 75 min in GM17 medium supplemented with 200 mM MgCl$_2$ and 2 mM CaCl$_2$. Chemically competent *E. coli* TOP10 (Invitrogen, Breda, The Netherlands) were used for transformation and amplification of the Pnis-dCas9-PAmCherry2-containing pUC16 plasmid (Supplementary Fig. 7).

Antibiotics were supplemented on agar plates to facilitate plasmid selection: 10 µg/ml chloramphenicol (for pTarget/pNonTarget) and 10 µg/ml erythromycin (for pLAB-dCas9). Screening for positive transformants was performed using colony PCR with KOD Hot Start Mastermix according to the manufacturer's instructions (Merck Millipore, Amsterdam, the Netherlands). Electrophoresis gels were made with 1% agarose (Eurogentec, Seraing, Belgium) in tris-acetate-EDTA (TAE) buffer (Invitrogen, Breda, The Netherlands). Plasmid digestions were compared with in silico predicted plasmid digestions (Benchling; https://benchling.com).

**pLAB-dCas9 plasmid construction.** Construction of the pLAB-dCas9 plasmid[41,61] was performed by synthesizing (Baseclear B.V., Leiden, The Netherlands) a codon-optimized fragment containing the sequence of Pnis-dCas9-PAmCherry2, flanked by XbaI/SalI restriction sites (Supplementary Fig. 7, Supplementary Note 1). This fragment was supplied in a pUC16 plasmid. After transformation in E. coli, the plasmid was isolated and digested with XbaI and SalI to obtain the Pnis-dCas9-PAmCherry2 fragment. From the pLABTarget expression vector[62], the Cas9 expression module was removed by digestion with XbaI and SalI, and replaced by the XbaI-SalI fragment containing Pnis-dCas9-PAmCherry2. The single-stranded guide RNA (sgRNA) for targeting pepN was constructed with the correct overhangs and inserted in the Eco31I digested sgRNA expression handle to yield the pLAB-dCas9 vector[62]. The plasmids used in this study, and vector maps for pLABTarget and pLAB-dCas9 are available upon request. pLAB-dCas9-PAmCherry2 was sequenced, and was confirmed to be intact in the used strains.

**pLAB-dCas9 no-sgRNA.** The pLAB-dCas9-nosgRNA plasmid was constructed by BoxI/SmaI digestion of the pLAB-dCas9-PAmCherry2 plasmid, and subsequent self-ligation. This resulted in deletion of the sgRNA handle and transcriptional terminator, successfully removing the functional sgRNA. The resulting pLAB-dCas9-nosgRNA plasmid was confirmed via sequencing.

**pTarget and pNonTarget plasmid construction.** The plasmid with binding sites for dCas9 (pTarget) was established by engineering five pepN target sites in the pNZ123 plasmid[63]. To this end, two single-stranded oligonucleotides (10 µl of 100 µM, each, Supplementary Table 3) that upon hybridization form the a single target sequence for the pepN-targeting sgRNA were incubated in 80 µl annealing buffer (10 mM Tris [pH = 8.0] and 50 mM NaCl) for 5 min at 100 °C, followed by gradual cooling to room temperature. The annealed mixed multiplexed oligonucleotides were cloned in HindIII-digested pNZ123. Afterwards, we selected a derivative that contains five pepN target sites via colony PCR (Supplementary Table 4). HindIII re-digestion was prevented by flanking the pepN DNA product by different base pairs, changing the HindIII site. Plasmids with five pepN target sites were designated pTarget (Supplementary Fig. 8). Plasmids without the pepN target sites (the original pNZ123 plasmids) were designated pNonTarget. The vector maps for pTarget and pNonTarget are shown in Supplementary Fig. 8. Correct insertion of the five pepN target sites was confirmed via sequencing.

**Construction of strains with pLAB-dCas9 and p(Non)Target.** Electro competent L. lactis NZ9000 cells[60] harbouring pLAB-dCas9 were transformed with pTarget or with pNonTarget and subsequently used for sptPALM or stored at −80 °C.

**Quantitative polymerase chain reaction (qPCR).** Both L. lactis strains containing pLAB-dCas9 and pTarget or pNonTarget were grown under similar lab conditions as the imaging experiments performed in this study (n = 2). After 3 h of growth, the cultures were split and dCas9 was induced (0 ng/ml nisin, 0,4 ng/ml nisin and 2 ng/ml nisin). The cells were then harvested after 12 h of growth by centrifugation. The cell pellets were washed, and DNA was extracted using InstaGene Matrix (Bio-Rad Laboratories, Richmond, California, USA).

Oligonucleotides were designed to amplify a region of spanning approximately 1000 base pairs on both pTarget and pNonTarget, and a region of similar length on the NZ9000 chromosome (Q3 + Q4 and Q7 + Q8; Supplementary Table 4). These oligonucleotides were used in a PCR reaction to generate templates which were diluted to function as a calibration curve in the following qPCR. Both qPCR reactions were performed on each isolated DNA sample (6 technical replicates) and the ratio between measured chromosomal amplicons (Q5 + Q6) and plasmid amplicons (Q1 + Q2) was determined. The samples which were uninduced with nisin were used to standardize the estimated pTarget and pNonTarget copy numbers.

**Sample preparation.** The strains to be used for single-molecule microscopy were grown o/n from glycerol stocks at 30 °C in chemically defined medium for prolonged cultivation (CDMPC)[64]. Then, they were sub-cultured at 5% v/v and grown for 3 h (average duplication time in CDMPC is ~90 min (determined via OD600 measurements)), before induced with 0.1 ng/ml nisin. 90 min later, the sample preparation began (see below).

Samples were prepared as described previously[41]. Briefly, after culturing of the cells, 0.5 µg/mL ciprofloxacin (Sigma-Aldrich, Zwijndrecht, The Netherlands) was added to slightly inhibit further cell division and DNA replication for sgRNA-pTarget and sgRNA-pNonTarget experiments[65]. Then, cells were centrifuged

(3500 RPM for 5 min; SW centrifuge (Froilabo, Mayzieu, France) with a CENSW12000024 swing-out rotor fitted with CENSW12000006 15 ml culture tube adaptors) and washed two times by gentle resuspension in 5 mL phosphate-buffered saline (PBS; Sigma-Aldrich, Zwijndrecht, The Netherlands). After removal of the supernatant, cells were resuspended in ~10–50 µL PBS from which 1–2 µL was immobilized on 1.5% 0.2 µm-filtered agarose (Certified Molecular Biology Agarose; BioRad, Veenendaal, The Netherlands) pads between two heat-treated glass coverslips (Paul Marienfeld GmbH & Co. KG, Lauda-Königshofen, Germany; #1.5H, 170 µm thickness). Heat treatment of glass coverslips involves heating the coverslips to 500 °C for 20 min in a muffle furnace to remove organic impurities.

**Experimental settings.** All imaging was performed on the miCube as described at 20 °C. A 561 nm laser with ~0.12 W/cm² power output was used for HiLo-to-TIRF illumination with 4 ms stroboscopic illumination[24] in the middle of 10 ms frames. Low-power UV illumination (µW/cm² range) was used and increased during experiments to ensure a low and steady number of fluorophores in the sample until exhaustion of the fluorophores. A UV-increment scheme was consistently used for all experiments (Supplementary Table 5). No emission filter was used except for the TIRF filter (Chroma ZET405/488/561m-TRF). The raw data were acquired using the open source Micro-Manager software[66]. During acquisition, 2 × 2 binning was used, which resulted in a pixel size of 128 × 128 nm. The camera image was cropped to the central 512 × 512 pixels (65.64 × 65.64 µm) or smaller. For sptPALM experiments, frames 500–55,000 were used for analysis, corresponding to 5–550 s. This prevented attempted localization of overlapping fluorophores at the beginning, and ensured a set end-time. 200–300 brightfield images were recorded by illuminating the sample at the same position as during the measurement. For the brightfield recording, we used a commercial LED light (INREDA, IKEA, Sweden) and a home-made diffuser from weighing paper.

**Localization.** To extract single molecule localizations, a 50-frame temporal median filter (https://github.com/marcelocordeiro/medianfilter-imagej) was used to correct background intensity from the movies[67]. In short, the temporal median filter determines the median pixel value over a sliding-window of 50 pixels to determine the median background intensity value for a pixel at a specific position and time point. This value is subtracted from the original data, and any negative values are set to 0. In the process, all pixels are scaled according to the mean intensity of each frame to account for shifts in overall intensity. The first and last 25 frames from every batch of 8096 frames are removed in this process.

Single particle localization was performed via the ImageJ[68]/Fiji[69] plugin ThunderSTORM[70] with added phasor-based single molecule localization algorithm (pSMLM[39]). Image filtering was done via a difference-of-Gaussians filter with Sigma1 = 2 px and Sigma2 = 8 px. The approximate localization of molecules was determined via a local maximum with a peak intensity threshold of 8, and 8-neighbourhood connectivity. Sub-pixel localization was done via phasor fitting[39] with a fit radius of 3 pixels (region-of-interest of 7-by-7 pixels). Custom-written MATLAB (The MathWorks, Natick, MA, USA) scripts were used to combine the output files from ThunderSTORM (Supplementary Software 1).

**Cell segmentation.** A cell-based segmentation on the localization positions was performed. First, a watershed was performed on the average of 300 brightfield-recorded frames of the cells. The watershed was done via the Interactive Watershed ImageJ plugin (http://imagej.net/Interactive_Watershed). Second, the localizations were filtered whether or not they fall in a pixel-accurate cell outline. If they do, a cell ID is added to the localization information.

**Estimating the copy number of dCas9.** The total copy number of dCas9 in a cell is not identical to the number of tracks found in each cell. Firstly, the UV illumination (405 nm wavelength) on the miCube required to photo-activate PAm-Cherry2 is not homogeneous over the complete field of view. To correct for this, a value for the average UV illumination experienced by each L. lactis cell is calculated. For this, a map of the UV intensity is made by placing a mirror on top of the objective and measuring the reflected scattering of the UV signal. Then, the mean UV intensity in the pixels corresponding to a cell according to the segmented brightfield images is stored. The cellular apparent dCas9 copy number is corrected for this normalized mean cellular UV intensity. Moreover, the cellular apparent dCas9 copy number was corrected for the average maturation grade of PAm-Cherry1, which is ~50%[53] (shown schematically in Supplementary Fig. 9). We assume the maturation grades of PAmCherry1 and PAmCherry2 to be similar.

**Tracking and fitting of diffusion coefficient histograms.** A tracking procedure was performed in MATLAB, using a modified Particle Point Analysis script[71] (https://nl.mathworks.com/matlabcentral/fileexchange/42573-particle-point-analysis) with a tracking window of 8 pixels (1.0 µm) and no memory (Supplementary Software 1). Localizations corresponding to different cells were excluded from being part of the same track. As the tracking window is of similar size as the cells itself, in practice all localizations in a cell are linked together in a track if they appear in successive frames.

An apparent diffusion coefficient, $D^*$, was then calculated for each track from the mean-squared displacement (MSD) of single-step intervals[72]. In short, for every

track with at least 4 localizations, the $D^*$ was calculated by calculating the mean square displacement between the first four steps and taking the average of that. Qualitative tracking information in cells (Fig. 2a, Fig. 3a) shows that diffusion coefficients up to ~4 $\mu m^2/s$ are obtained. These high diffusion coefficient tracks are caused by including false positive localizations in tracks. Therefore, tracks with a diffusion coefficient clearly caused by inclusion of false positive localizations ($D^* > 2.5\ \mu m^2/s$) were excluded from further analysis. We then binned the diffusion coefficients in 40 logarithmic-divided bins from $D^* = 0.01$ to $D^* = 2.5\ \mu m^2/s$. The pNonTarget diffusional information was first corrected for the diffusion histogram obtained from a non-induced sample, subtracting the non-induced histogram from the pNonTarget histogram based on the approximated relative size of the non-induced histogram (~7.2 tracks per cell were found in non-induced cells).

Then, a Monte-Carlo diffusion distribution analysis (MC-DDA; described below) consisting of 20.000 dCas9 proteins was fitted via a general Levenberg-Marquardt fitting procedure in MATLAB. The error of this fit was determined via a general bootstrapping approach, where a $D^*$-list with the same length as the original, but randomly filled with values from the original (allowing for more than one entry of the same data), was fitted via the same procedure. For the pTarget diffusional information, the pNonTarget best fitted model (calculated via the same model, but with 100.000 dCas9 proteins) was fitted and smoothed via a Savitzky-Golay filter with order 3 and length 7, to reduce noise on the fit, alongside a single population following the following equation:

$$y = \frac{\left(\frac{n}{D_{plasmid}}\right)^n \cdot x^{(n-1)} \cdot e^{-n\frac{x}{D_{plasmid}}}}{(n-1)!} \tag{1}$$

Where $D_{plasmid}$ is the $D^*$-value corresponding to plasmid-bound dCas9, $n$ the number of steps in the trajectory (set to four in this study), $y$ the count of the histogram, and $x$ the $D^*$-value of the histogram. $D_{plasmid}$ was kept constant in the global fit, while the size of this population and the size of the pNonTarget model were allowed to vary between apparent cellular dCas9 copy number bins. Again, the error of this fit was determined via a general bootstrapping approach.

**pNonTarget Monte-Carlo diffusion distribution analysis**. The pNonTarget data is fitted with a Monte-Carlo diffusion distribution analysis (MC-DDA), in which a variable $D_{free}$, localization error, $k_{free \rightarrow bound}$, and $k_{bound \rightarrow free}$ need to be provided (Supplementary Software 1). A set number of dCas9 proteins are simulated (20,000 for the fit, 100,000 for visualising the fit). These proteins are then randomly placed in a cell, which is simulated as a cylinder with length 0.5 $\mu m$ and radius 0.5 $\mu m$, capped by two half-spheres with radius 0.5 $\mu m$, and the current state of the proteins is set to free or immobile, based on the respective kinetic rates ($c_{bound} = k_{free \rightarrow bound}$ /($k_{bound \rightarrow free} + k_{free \rightarrow bound}$), $c_{free} = 1 - c_{bound}$). Moreover, the proteins are given a time before they are changed between states (log(rand)/$-k$, where rand is a random number between 0 and 1, and $k$ is the respective kinetic rate). Then, the movement of the proteins is simulated with over-sampling with regards to the frame time (0.1 ms). The free proteins will move a distance equal to a randomly sampled normal distribution with $\sigma = \sqrt{2 \cdot D_{free} \cdot steptime}$, where steptime is 0.1 ms. Then, it checked if this position is still within the cell outline. If not, a new location will be pulled from the distribution and checked against the cell outline. Every time-step, the time until state-change is subtracted with the time-step, and if this value becomes $\leq 0$, the proteins will switch states, getting a new diffusion coefficient and state-change time. Every 10 ms after an initial equilibrium time of 200 ms, the current location of the proteins is convoluted with a random localization error, from a randomly sampled normal distribution with $\sigma = $ localization error. The simulation is ended after 5 localization points are acquired for every protein. Further tracking and diffusion coefficient calculations are done the same as the experimental data.

**Target simulation**. For the target simulation, a certain number of dCas9 are simulated (similar to the average of the bins used in experiments), alongside a variable total number of PAM sites (1/16 chance at ~7.5 mln bases, or 1.5× double-stranded *L. lactis* genome[73]), plasmid copy number, target sites (5 per plasmid), incubation time (90 min), fluorophore maturation time (20 min[27]), and a $k_{off}$ rate (Supplementary Software 1). The dCas9 proteins are simulated one by one. The first dCas9 will have access to all target sites, and will be simulated for [incubation time], assuming the first dCas9 was made exactly at the start of the nisin incubation. Subsequent dCas9 proteins will have access to fewer target sites, depending on whether or not earlier dCas9 proteins have ended the simulation bound to target sites. Subsequent dCas9 proteins will also be simulated for a shorter time, linearly scaling from [incubation time] to [fluorophore maturation time], which assumes that dCas9 proteins are steadily produced throughout the incubation time, but allowing for the fact that dCas9 proteins that do not yet have a matured PAmCherry2 are not visible during sptPALM.

Then, the dCas9 proteins randomly start in the free, PAM-probing, or target-bound state, based on the previously determined kinetic constants, similarly as in the pNonTarget simulation. The proteins are also given a time until state change, as was done in the pNonTarget simulation. Next, the simulation time of a single dCas9 protein was decreased by this time until state change, whereupon a new state was given to the protein: free proteins changed to PAM-probing or target-bound,

with the target-bound chance being equal to $^{nr\ target\ sites}/_{total\ nr\ of\ PAM\ sites}$; PAM-probing or target-bound proteins were changed to free proteins. This was continued until the end of the simulation, after which the final state was determined. If the dCas9 was bound to a target, the available target sites were decreased by 1 for the next simulated dCas9. The reported values are the mean of 50 repetitions of the simulation, with the 95% confidence interval determined via the standard deviation of these repetitions.

For simulating Cas9 cleavage rates, it was assumed that a single target site was present and that a dCas9 would never be removed from a target site. By then analysing the bound dCas9, it indicates whether the target site has been cleaved by Cas9. The other simulation parameters were kept constant.

**miCube drift quantification**. We characterised the positional stability of the miCube via super-resolution measurements of GATTA-PAINT 80R DNA-PAINT nanorulers (GATTAquant GmbH, Germany). We imaged the nanorulers in total internal reflection (TIR) mode using a 561 nm laser (~7 mW) with a frame time of 50 ms using 2 × 2 pixel binning on the Andor Zyla 4.2 PLUS sCMOS. Astigmatism was enabled by placing a 1000 mm focal length astigmatic lens (Thorlabs) 51 mm away from the camera chip. A video of 10,000 frames was recorded via the MicroManager software[66].

After recording the movie, we first localized the $x$, $y$, and $z$-positions of the point spread functions of excited DNA-PAINT nanoruler fluorophores with the ThunderSTORM software[70] for ImageJ[68] with the phasor-based single molecule localization (pSMLM) add-on[39]. The ThunderSTORM software was used with the standard settings, and a 7 by 7 pixel region of interest around the approximate centre of the point spread functions was used for pSMLM. To determine the $z$-position, we compared the astigmatism of the point-spread function to a pre-recorded calibration curve recorded using immobilized fluorescent latex beads (560 nm emission peak, 50 nm diameter).

After data analysis, we performed drift-correction in the lateral plane using the cross-correlation method of the ThunderSTORM software. The cross-correlation images were calculated using 10x magnified super-resolution images from a sub-stack of 1000 original frames. The fit of the cross-correlation was used as drift of the lateral plane. Drift of the axial plane was analysed by taking the average $z$-position of all fluorophores, assuming that all DNA-PAINT nanorulers are fixed to the bottom of the glass slide.

**Reporting summary**. Further information on research design is available in the Nature Research Reporting Summary linked to this article.

## Data availability
The source data underlying Figs. 2 and 3 and Supplementary Figs. 4–6 and 8 are provided as a Source Data file.

## Code availability
All code necessary to perform this study is made available as Supplementary Information (Supplementary Software 1, which further contains in accompanying programming flowchart).

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

## Acknowledgements

K.J.A.M. is funded by a VLAG PhD-fellowship grant awarded to J.H. J.H. acknowledges funding from the Innovation Program Microbiology Wageningen (IPM-3). S.v.d.E is funded by the BE-Basic R&D program, which was granted a FES subsidy from the Dutch Ministry of Economic affairs. The authors thank the WOSM (Warwick Open Source Microscope, see www.wosmic.org) for inspiration.

## Author contributions

K.J.A.M., S.B., and J.H. designed, built and characterised the miCube setup. K.J.A.M., S.P.B.v.B. and G.A.V. recorded and analysed the experimental single molecule data. J.H., S.v.d.E. and P.v.B. envisioned using *L. lactis*, dCas9, fluorescent proteins and p(Non-)Target cells to conduct super-resolution single molecule studies. S.P.B.v.B., S.v.d.E., P.v.B., and M.K. designed the DNA vectors used in this study. S.P.B.v.B. and S.v.d.E. assembled the DNA vectors and transformed cells. K.J.A.M., J.N.A.V., and J.H. developed DDA. K.J.A.M. and J.N.A.V. wrote software for data analysis. J.N.A.V. and S.J.J.B. provided reagents and expertise for setting up the single molecule assays. K.J.A.M. and J.H. wrote the manuscript with input from all authors. J.H. initialised the study and the collaborations, and supervised all aspects of the study.

## Additional information

**Competing interests:** The authors declare no competing interests.

