## [Peer Review File · Nature Communications]

Editorial Note: This manuscript has been previously reviewed at another journal that is not operating a transparent peer review scheme. This document only contains reviewer comments and rebuttal letters for versions considered at Nature Communications. Mentions of the other journal have been redacted.

Reviewers' comments:

Reviewer #1 (Remarks to the Author):

This manuscript is a re-submission of a manuscript previously submitted to [redacted]. Overall, the authors have made a serious attempt at addressing my comments. In general, I am happy with the revisions. The MC-DDA fitting is definitely an improvement and avoids the problem of assigning the populations. I also find that the new data in the absence of guide is important. However, I still find that there are some issues with the distributions. There still appears to be an unexpected bound population present in either the pNonTarget or the no-sgRNA experiments. This is particularly prominent in the low copy number pNonTarget (Sup Fig 4a. left) and high copy number no-sgRNA pTarget (Sup Fig 5 bottom right). The authors should expand the discussion to provide explanations for the presence of these populations. The reanalysis of the changes with increased copy number addresses our concern about the claim that of a decrease in the Dbound population is observed with increased copy number. As the differences in bound and unbound populations in all experiments are more convincing with the low copy numbers, it might be clearer to not include the high copy number experiments. Another issue is how the distributions of the no-sgRNA pTarget experiments (Sup Fig 5 bottom) are carried out. There are two populations as indicated by the mismatch between the experimental data and the fit and the non-randomly distributed residuals (particularly for high copy number distributions). These fits need to be re-done.

Reviewer #3 (Remarks to the Author):

All technical concerns are addressed and the manuscript seems ready for publication.

Reviewer #1 (Remarks to the Author):

This manuscript is a re-submission of a manuscript previously submitted to [redacted]. Overall, the authors have made a serious attempt at addressing my comments. In general, I am happy with the revisions.

Our response: We thank reviewer #1 for his/her careful examination of our revised manuscript.

The MC-DDA fitting is definitely an improvement and avoids the problem of assigning the populations. I also find that the new data in the absence of guide is important. However, I still find that there are some issues with the distributions.

There still appears to be an unexpected bound population present in either the pNonTarget or the no-sgRNA experiments. This is particularly prominent in the low copy number pNonTarget (Sup Fig 4a. left) and high copy number no-sgRNA pTarget (Sup Fig 5 bottom right). The authors should expand the discussion to provide explanations for the presence of these populations.

Our response: We cannot find indications of an additional unaccounted bound population in figure SFig. 4a (same as Fig.2b). The experimental data is fit very well with the two-state model (MC-DDA), and the residuals do not show any systematic deviation. As described in our manuscript (Results: “dCas9 is PAM-screening for 17 ms”), the bound fraction (centred around $\sim 0.15\text{-}0.2 \mu\text{m}^2/\text{s}$) represents situations in which the PAM screening (an intrinsic property of Cas9) leads to dCas9 bound to DNA for durations of comparable lengths as the track (10-40 ms).

The reanalysis of the changes with increased copy number addresses our concern about the claim that of a decrease in the Dbound population is observed with increased copy number. As the differences in bound and unbound populations in all experiments are more convincing with the low copy numbers, it might be clearer to not include the high copy number experiments.

Our response: We agree with the reviewer that the target bound population is highest with low protein copy numbers. However, for building our model of dCas9 interference (Fig.3c leading to Fig.4), the high copy number data is essential, and we do not want to omit it.

Another issue is how the distributions of the no-sgRNA pTarget experiments (Sup Fig 5 bottom) are carried out. There are two populations as indicated by the mismatch between the experimental data and the fit and the non-randomly distributed residuals (particularly for high copy number distributions). These fits need to be re-done.

Our response: We thank the reviewer for pointing out the mismatch. We analysed the no-sgRNA nNonTarget data and then used the obtained rates to obtain a target-bound (pTarget), identically to the ‘normal’ pTarget data fitting procedure. As seen in SFig.5 bottom (now SFig.5c), no target bound fractions could be found. Encouraged by the reviewer, we decided to add an additional row, showing the direct two-state fitting of the no-sgRNA pTarget data (now SFig.5b). As expected, the obtained rates are similar to the corresponding no-sgRNA pNonTarget data, and thus also similar to ‘normal’ pNonTarget data.

Reviewer #3 (Remarks to the Author):

All technical concerns are addressed and the manuscript seems ready for publication.

Our response: We thank reviewer #3 for his/her positive evaluation of our manuscript.